# Aligning CLIP Visual Features with Object Masks via Spatial Remapping for Open-Vocabulary Segmentation

## Abstract

Open-vocabulary panoptic segmentation aims to segment and categorize objects from novel classes. Existing methods typically follow a two-stage process: first generating object masks, then classifying objects by comparing text embeddings with object embeddings obtained via mask pooling on CLIP visual features. However, because CLIP was pretrained without object mask supervision, its visual features poorly align with object masks, which impedes accurate category matching. To tackle this challenge, certain approaches ensemble object embeddings from CLIP visual features with those from newly learned segmentation features. However, because the newly learned features lack pretraining on large-scale datasets, they generalize poorly to unseen categories. To address these limitations, we introduce a spatial remapping module for CLIP, tailored for open-vocabulary segmentation. This module spatially remaps CLIP visual features to better align with object masks by leveraging spatial relationships between CLIP visual and segmentation features. Consequently, object embeddings obtained via mask pooling on the remapped CLIP visual features correctly match the text embeddings. This approach eliminates the need for mask pooling on newly learned features, thereby better preserving CLIP's zero-shot image-text alignment ability. To improve efficiency and accuracy, we model these spatial relationships at the object level and use object mask annotations for supervision. Our model, trained on the COCO panoptic dataset, demonstrates exceptional zero-shot performance across various datasets, affirming its effectiveness.

## 1 Introduction

Panoptic segmentation (Kirillov et al., 2019) is a fundamental vision task aiming at classifying each pixel in an image into different instances and categories. It melds the intricacies of both instance and semantic segmentation, playing an important role in numerous fields, such as autonomous driving (Zendel et al., 2022), robotics (Mohan & Valada, 2022), and augmented reality (Dahnert et al., 2021). Recently, this field has rapidly shifted to open-world settings, targeting segmenting and categorizing objects from novel classes, termed as open-vocabulary panoptic segmentation.

Current methods in open-vocabulary panoptic segmentation (Liang et al., 2023; Ding et al., 2023; Xu et al., 2023d; Qin et al., 2023; Xu et al., 2023a; Yu et al., 2023a; Wang et al., 2024) typically employ a two-stage pipeline to leverage CLIP's (Radford et al., 2021) zero-shot image-text alignment capability. First, they generate object masks using a class-agnostic mask proposal network. Then, they classify the objects by comparing text embeddings encoded from class names with object embeddings obtained through mask pooling on CLIP visual features, as depicted in Fig. 2 (a). However, because CLIP was pretrained on image-level captions without object mask supervision, its visual features do not align well with object masks, as shown in Fig. 1 (c). This misalignment impedes the accuracy of matching text embeddings with object embeddings obtained from mask pooling, as depicted in Fig. 1 (d).

To address this problem, certain approaches (Yu et al., 2023a; Xu et al., 2023a; Wang et al., 2024) ensemble object embeddings from CLIP visual features with those from newly learned visual features, as illustrated in Fig. 2 (b). Although these newly learned visual features align well with object masks due to segmentation

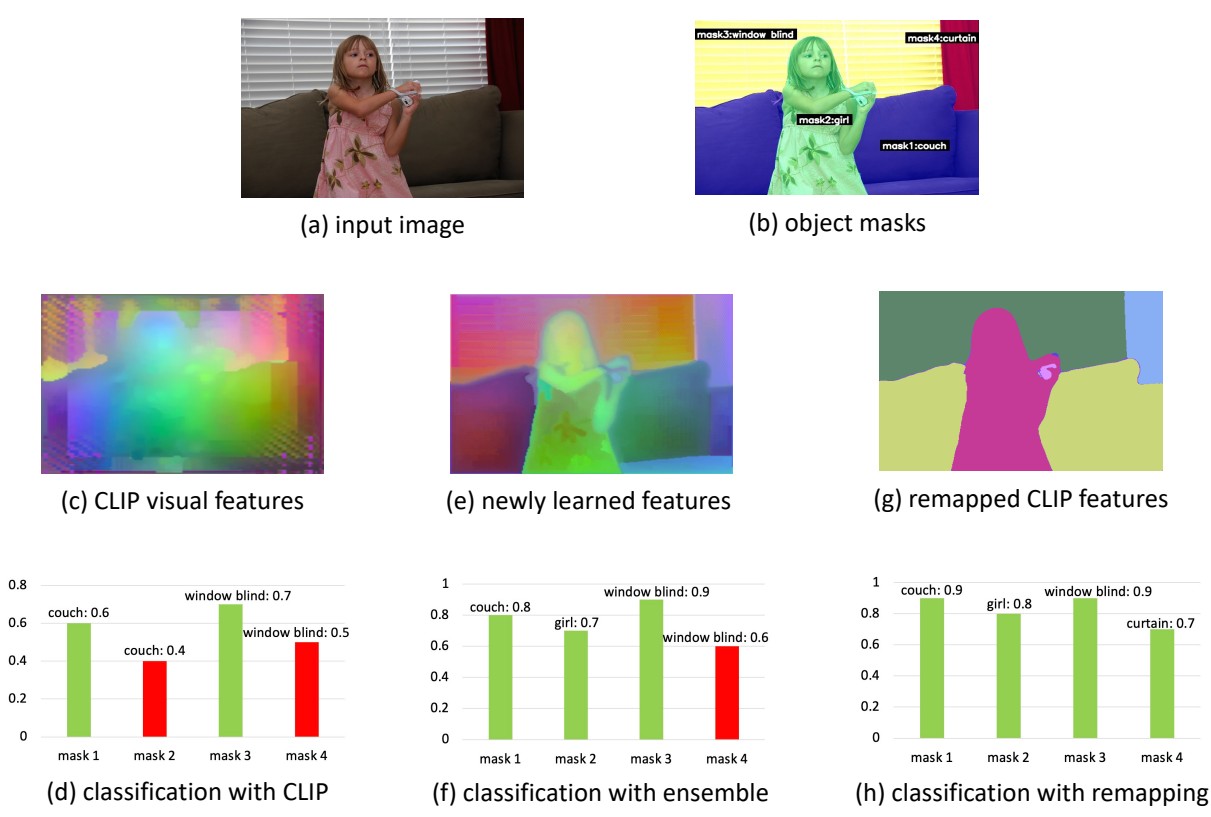

Figure 1: Comparison of object classification using different features in mask pooling. (a) input image; (b) object masks; (c) t-SNE visualization of CLIP visual features; (d) classification results using CLIP visual features; (e) t-SNE visualization of newly learned visual features; (f) classification results using an ensemble of CLIP and newly learned visual features; (g) t-SNE visualization of remapped CLIP visual features, and (h) classification results using remapped CLIP visual features.

training (as shown in Fig. 1 (e)), the object embeddings derived from them generalize poorly to unseen categories because of the limited diversity of categories in the training datasets. Consequently, the ensembled object embeddings still cannot accurately match the text embeddings during zero-shot object classification (as shown in Fig. 1 (f)).

To overcome these limitations, we propose a spatial remapping module for CLIP, specifically designed for open-vocabulary segmentation. As illustrated in Fig. 2 (c), this module spatially remaps CLIP visual features to better align with object masks, by leveraging the spatial relationships between CLIP visual and segmentation features. Subsequently, object embeddings are obtained via mask pooling on the remapped CLIP visual features. This process eliminates the need to obtain object embeddings by mask pooling on newly learned visual features for alignment with object masks, thereby preserving CLIP's zero-shot image-text alignment ability as much as possible. As shown in Fig. 1 (g), after the spatial remapping, the CLIP visual features are precisely aligned with object masks. Consequently, the object embeddings, derived from mask pooling on the remapped CLIP visual features, correctly match the text embeddings, enabling accurate object classification, as depicted in Fig. 1 (h).

Learning the spatial relationships between CLIP visual features and segmentation features in pairs is computationally infeasible due to its $O(N^2)$ complexity. To reduce computational and memory costs, we model the spatial relationships at the object level by decomposing them into two components: (1) the relationships between segmentation features and object features (i.e., predicted object masks), and (2) the relationships between CLIP visual features and object features. This approach enables us to use object mask annotations

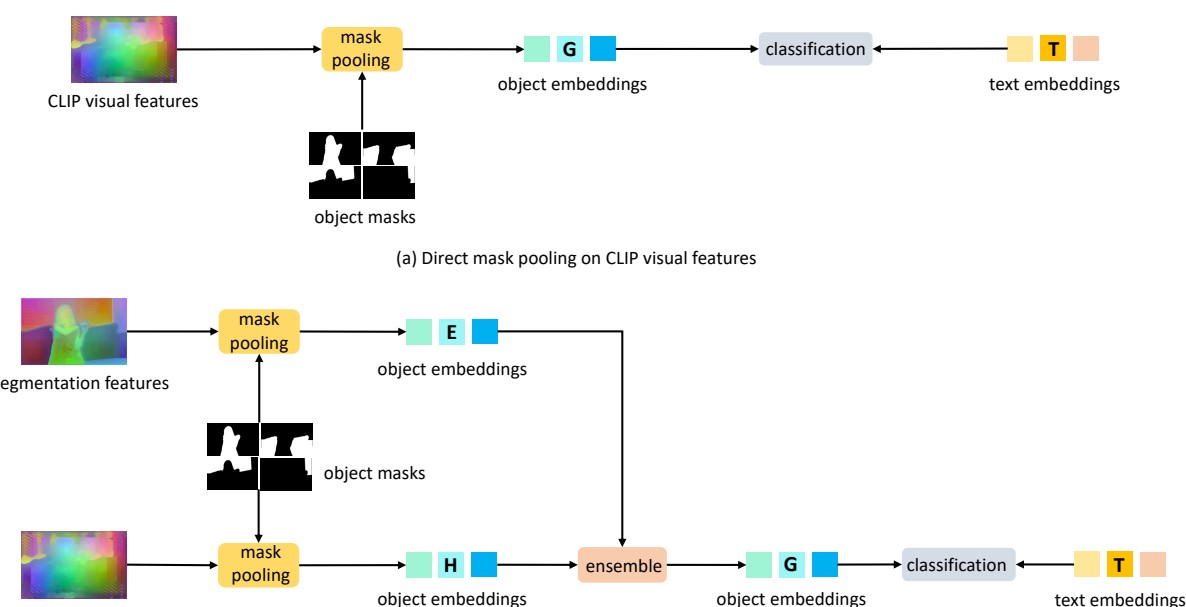

(a) Direct mask pooling on CLIP visual features

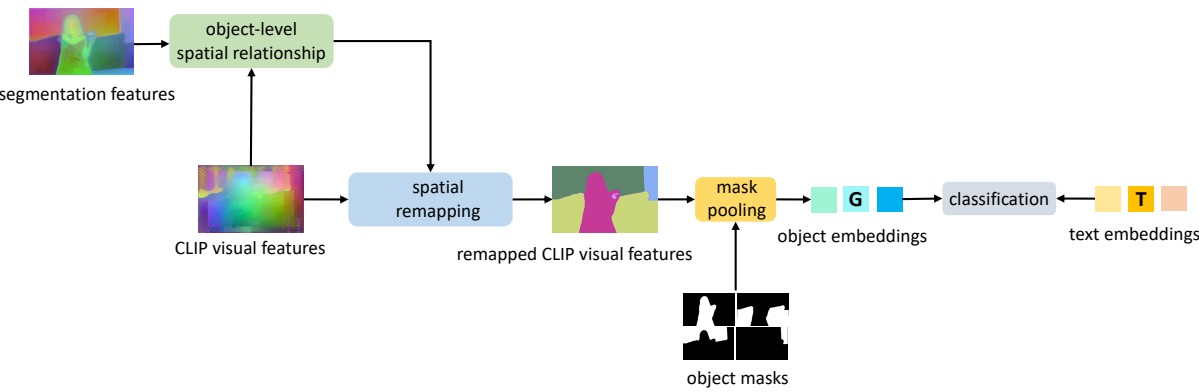

(b) Ensemble embeddings from CLIP visual features and segmentation features

(c) Spatial Remapping of CLIP visual features

Figure 2: The comparison of the pipelines for open-vocabulary segmentation. (a) Vanilla pipeline, extracting object embeddings via mask pooling on CLIP visual features. (b) Ensemble pipeline, ensembling object embeddings from CLIP visual features with those from newly learned visual features. (c) Proposed pipeline, spatially remapping CLIP visual features before applying mask pooling to extract object embeddings

to supervise the decomposed relationships, thereby improving accuracy. Following (Yu et al., 2023a), we derive segmentation features from a pixel decoder appended to the CLIP visual backbone.

The main contributions of our work are three-fold:

- We propose a spatial remapping module for CLIP that aligns CLIP visual features with object masks, enhancing open-vocabulary segmentation.

- We model the spatial relationships between CLIP visual features and segmentation features at the object level, which enhances both computational efficiency and accuracy.

- Trained on the COCO panoptic dataset and validated on other datasets such as Cityscapes, ADE20K, Mapillary Vistas, Pascal VOC and Pascal Context, our model demonstrates state-of-the-art performance in open-vocabulary segmentation.

## 2 Related Work

### 2.1 Vision-Language Pre-training

Vision-Language Pre-training (VLP) learns joint representations from visual and textual data, bridging the gap between both modalities. CLIP (Radford et al., 2021) pioneers this approach by training on vast image-text pairs from the internet, facilitating zero-shot transfer across diverse tasks like image classification and retrieval. BLIP (Li et al., 2022b) enhances CLIP with the Multimodal Mixture of Encoder-Decoder (MED) architecture, bootstrapping noisy web data for improved understanding and generation. BLIP-2 (Li et al., 2023b) further optimizes computation using frozen pre-trained unimodal models, bridged by a Querying Transformer. Building on BLIP-2, InstructBLIP (Dai et al., 2023) introduces instruction tuning, transforming multiple datasets into an instruction format and achieving state-of-the-art performance. Recent vision-language models have evolved to Multi-Modal Large Language Models (MLLMs), which excel at integrating and processing information across modalities like text, images, audio, and video. Recent advancements in MLLMs focus on instruction tuning and generative modeling for richer and deeper multi-modal comprehension. For example, LLaVA (Liu et al., 2024a) integrates vision encoders into LLMs, then fine-tunes them on multi-turn instruction datasets to enhance visual understanding. GPT-4V(ision) (Achiam et al., 2023) integrates vision capabilities into an LLM framework, enabling open-ended reasoning and description generation. These models excel in multi-modal dialogue, visual question answering, and image captioning.

The aforementioned works primarily focus on global representations tailored for image classification, but there is a burgeoning interest in object-level representations for detection and segmentation. For instance, RegionCLIP (Zhong et al., 2022) adapts the CLIP model to learn region-level representations and addresses the domain shift by aligning image regions with texts during pre-training, excelling in open-vocabulary object detection. Grounding DINO (Liu et al., 2023) integrates Transformer-based detection with grounded pre-training, enabling detection of arbitrary objects using human inputs like category names. However, open-vocabulary segmentation still faces challenges due to the lack of large-scale mask-text paired datasets.

### 2.2 Open Vocabulary Semantic Segmentation

Open-vocabulary semantic segmentation (OVSS) aims to assign pixel-wise labels from an arbitrary vocabulary without instance separation. Early OVSS methods established the recipe of aligning dense visual features with text embeddings and performing per-pixel classification with a free vocabulary. Representative baselines include LSeg (Li et al., 2022a), which learns pixel-level visual–text alignment and enables zero-shot labeling by arbitrary text queries, and OpenSeg (Ghiasi et al., 2022), which scales OVSS with image-level captions by first proposing masks and then aligning mask features with words. Subsequent generalist decoders (e.g., X-Decoder (Zou et al., 2023)) further unify pixel outputs and language tokens in a shared space, demonstrating strong transfer to OVSS benchmarks. Side-Adapter designs (SAN) attach lightweight branches to frozen CLIP to produce CLIP-aware proposals for semantic labeling, offering a practical accuracy–efficiency trade-off (Xu et al., 2023b). More recent CLIP-centric baselines and their mask-adapted variants (e.g., SegCLIP (Luo et al., 2023)) continue to validate dense alignment as the core of OVSS.

Recent OVSS work strengthens *semantic alignment and calibration*. SCAN (Liu et al., 2024b) explicitly calibrates the in-/out-vocabulary embedding space by injecting CLIP's semantic priors and a contextual-shift strategy, consistently improving performance on standard OVSS benchmarks. EBSeg (Shan et al., 2024) proposes image-embedding balancing (AdaB decoder + semantic-structure consistency) to mitigate bias toward seen classes, yielding strong gains on ADE20K/COCO-Stuff, etc. A complementary line focuses on *parameter-efficient or controllable fine-tuning*. H-CLIP (Peng et al., 2025a) performs symmetric PEFT in hyperspherical space on both CLIP modalities, reducing compute while preserving generalization and improving unseen-class segmentation. HyperCLIP (Peng et al., 2025b) also shows that carefully (co-)tuning text and image encoders—rather than freezing one branch—can further enhance pixel-level open-set recogni-

tion, offering guidance on what to tune and where to regularize. Beyond vanilla CLIP tuning, dual-semantic guidance (Wang et al., 2025) explicitly refines both visual and textual semantics to reduce background interference and enhance fine-grained recognition. New prompting-based designs (e.g., dual-prompt cost-volume learning (Zhao et al., 2025)) inject shallow-feature cues for small/fine structures while narrowing the image–text domain gap at the pixel level.

## 2.3 Open Vocabulary Panoptic Segmentation

Open vocabulary panoptic segmentation aims to segment and recognize objects of arbitrary categories. Existing methods usually adopt a two-stage pipeline. For example, FreeSeg (Qin et al., 2023) first extracts object masks from input images, and then applies CLIP to classify these masks by matching object embeddings, encoded from the masked images, with text embeddings encoded from class names. OVSeg (Liang et al., 2023) further improves CLIP's zero-shot object classification abilities by finetuning it on a collection of masked image regions and their respective text descriptions. ODISE (Xu et al., 2023a) utilizes a pre-trained text-to-image diffusion model to generate object masks and their embeddings. These embeddings are combined with those obtained from the CLIP feature map through mask pooling for object classification. FC-CLIP (Yu et al., 2023a) streamlines this by directly employing a shared frozen convolutional CLIP backbone along with a multi-scale pixel decoder for object mask generation. Object embeddings for classification are obtained by applying mask pooling to both the decoder's output feature map and the original CLIP feature map.

MaskCLIP (Ding et al., 2023) integrates mask class tokens with a pre-trained ViT CLIP model, using object masks as attention masks, to obtain object embeddings for classification. MasQCLIP (Xu et al., 2023d) further enhances MaskCLIP with mask progressive distillation and class token fine-tuning. MaskCLIP++ (Zeng et al., 2024) observes that low-quality pseudo-masks can harm regional alignment; thus, it fine-tunes CLIP with ground-truth masks plus a consistency constraint, improving mask classification and boosting several benchmarks. MAFT-Plus (Jiao et al., 2024) optimizes collaborative vision–text representations and reports consistent improvements across standard benchmarks without requiring a separately fine-tuned CLIP-V branch. Despite these advancements, these methods are still hindered by limited training data and a non-negligible domain gap between the obtained object embeddings and the original CLIP image embeddings.

## 2.4 Segmentation Foundation Models

Segmentation foundation models are designed to execute a variety of segmentation tasks under the guidance of user prompts, showcasing robust capabilities in zero-shot segmentation. A notable contribution in this field is the Segment Anything Model (SAM) (Kirillov et al., 2023), which is trained on the large-scale SA-1B dataset in a promptable manner, enabling zero-shot transfer to new image distributions and tasks, such as interactive tracking and video segmentation (Yang et al., 2023) and mask-free image inpainting (Yu et al., 2023b). More recently, SAM-2 (Ravi et al., 2024) extends promptable segmentation from images to images and videos with a simple transformer equipped with streaming memory, enabling real-time interactive segmentation and long-range temporal association. SAM-2 is trained with a model-in-the-loop data engine that bootstraps annotations to build SA-V, currently one of the largest video segmentation datasets; the memory design reduces user interaction and improves zero-shot transfer across domains. Very recently, SAM-3 introduces Promptable Concept Segmentation (PCS), where concept prompts—short noun phrases, image exemplars, or their combination—return segmentation masks and unique identities for all matching instances. SAM-3 uses a shared vision backbone for an image-level detector and a memory-based video tracker. It decouples recognition from localization via a presence head; paired with a scalable data engine that mines approximately 4 million unique concept labels with hard negatives across images and videos, this design markedly improves concept coverage.

While SAM is versatile, it struggles with intricate object structures and some task-specific data. To overcome these, HQ-SAM (Ke et al., 2023) introduces a high-quality output token and global-local feature fusion for improved mask prediction, while SAM-Adapter (Chen et al., 2023a) incorporates domain-specific information using adapters to improve performance in specific tasks. However, these methods neglect the importance of

object semantics for zero-shot segmentation. To further equip SAM with semantic awareness, Semantic-SAM (Li et al., 2023a) consolidates multiple datasets and train on decoupled objects and parts classification, and SAM-CLIP (Wang et al., 2023) merges the capabilities of CLIP and SAM using multi-task distillation and continual learning. However, due to the lack of a large-scale mask-text paired dataset, the open-vocabulary object recognition performance of these methods remains unsatisfactory. Furthermore, the robustness of their zero-shot segmentation performance is compromised as it depends on bounding box prompts from an external object detector.

## 3 Method

In this section, we present the proposed method for open-vocabulary segmentation. We first present the overall framework, and then describe each component in detail, including a spatial remapping module, an object-level spatial relationship module, and training losses.

### 3.1 Overall Framework

The framework, illustrated in Fig. 3 (a), begins by extracting CLIP visual features $\mathbf{F}$ from the input image $\mathbf{I}$ using a frozen CLIP visual encoder. A pixel decoder is appended to this encoder to obtain segmentation features $\mathbf{B}$. An object-level spatial relationship module is then applied to establish spatial relationships between the CLIP visual features $\mathbf{F}$ and the segmentation features $\mathbf{B}$. Leveraging these relationships, a spatial remapping module transforms the CLIP visual features $\mathbf{F}$ into a new feature map $\mathbf{V}$ that better aligns with the object masks. Object embeddings $\mathbf{G}$ are derived by applying mask pooling to the remapped feature map $\mathbf{V}$, and these embeddings are subsequently matched with text embeddings $\mathbf{T}$ from a frozen CLIP text encoder for object classification.

### 3.2 Object-Level Spatial Relationship Module

In this section, we introduce the proposed object-level spatial relationship module, which establishes spatial relationships between the CLIP visual features and the segmentation features. These relationships are utilized for the spatial remapping of the CLIP visual features.

As illustrated in Fig. 3 (a), given an input image $\mathbf{I} \in \mathbb{R}^{H_0 \times W_0 \times 3}$, we first apply a frozen CLIP visual encoder to extract CLIP visual features $\mathbf{F} \in \mathbb{R}^{H \times W \times C}$. Following (Yu et al., 2023a), we then append a pixel decoder to the CLIP visual encoder to obtain segmentation features $\mathbf{B} \in \mathbb{R}^{H \times W \times C}$. Here, $H$, $W$, and $C$ denote the height, width, and number of channels of the feature maps $\mathbf{F}$ and $\mathbf{B}$, respectively.

To reduce computational and memory costs, we model these spatial relationships at the object level, by utilizing object features as intermediaries, as shown in Fig. 3 (b). Specifically, we initialize a set of learnable object queries $\mathbf{Q} \in \mathbb{R}^{L \times C}$, where $L$ is the number of object queries. The object features $\mathbf{O} \in \mathbb{R}^{L \times C}$ are then extracted from the segmentation features $\mathbf{B}$ and the CLIP visual features $\mathbf{F}$ through two cross-attentions as follows:

$$
\begin{aligned}
\mathbf{O}'_i &= \mathbf{Q}_i + \sum_{j=1}^{H} \sum_{k=1}^{W} f_{\mathrm{F}}(\mathbf{Q}_i, \mathbf{F}_{jk}) g_{\mathrm{F}}(\mathbf{F}_{jk}), \\
\mathbf{O}_i &= \mathbf{O}'_i + \sum_{j=1}^{H} \sum_{k=1}^{W} f_{\mathrm{B}}(\mathbf{O}'_i, \mathbf{B}_{jk}) g_{\mathrm{B}}(\mathbf{B}_{jk}),
\end{aligned}
\tag{1}
$$

where $\mathbf{O}_i \in \mathbb{R}^C$ denotes the $i$-th object feature corresponding to the $i$-th object query $\mathbf{Q}_i$; $\mathbf{F}_{jk}$ and $\mathbf{B}_{jk} \in \mathbb{R}^C$ are the features at the $j$-th row and $k$-th column of the feature maps $\mathbf{F}$ and $\mathbf{B}$, respectively; $g_{\mathrm{F}}(\mathbf{F}_{jk}) = \mathbf{W}_g \mathbf{F}_{jk}$ is a linear transformation function, where $\mathbf{W}_g \in \mathbb{R}^{C \times C}$ is a learnable weight matrix; $f_{\mathrm{F}}(\mathbf{Q}_i, \mathbf{F}_{jk})$ computes a similarity between $\mathbf{Q}_i$ and $\mathbf{F}_{jk}$ as follows:

$$
f_{\mathrm{F}}(\mathbf{Q}_i, \mathbf{F}_{jk}) = \frac{\exp(\mathbf{Q}_i^\top \mathbf{W}_\theta^\top \mathbf{W}_\phi \mathbf{F}_{jk})}{\sum_{j'=1}^{H} \sum_{k'=1}^{W} \exp(\mathbf{Q}_i^\top \mathbf{W}_\theta^\top \mathbf{W}_\phi \mathbf{F}_{j'k'})},
\tag{2}
$$

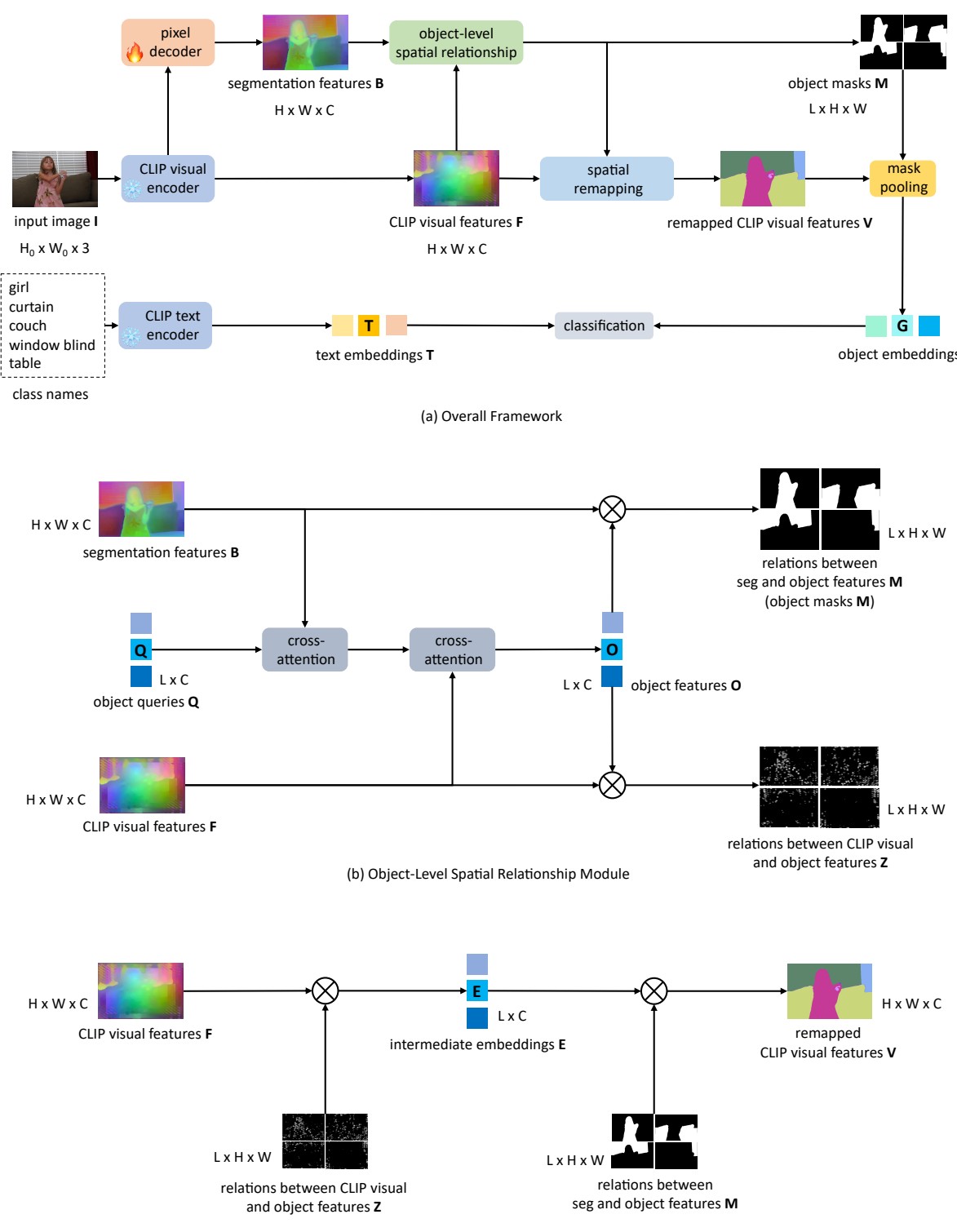

Figure 3: Overview of the proposed framework. It includes a frozen CLIP visual encoder to extract visual features, a pixel decoder to generate segmentation features, an object-level spatial relationship module to model spatial relationships between these features, and a spatial remapping module that remaps the CLIP visual features based on these relationships to achieve alignment with object masks.

where $\mathbf{W}_\theta$ and $\mathbf{W}_\phi \in \mathbb{R}^{C \times C}$ are learnable weight matrices; $g_{\mathrm{B}}(\cdot)$ and $f_{\mathrm{B}}(\cdot, \cdot)$ have the same form as $g_{\mathrm{F}}(\cdot)$ and $f_{\mathrm{F}}(\cdot, \cdot)$, respectively, except for the learnable weight matrices. Next, we calculate the relationships between the CLIP visual features $\mathbf{F}$ and the object features $\mathbf{O}$, denoted as $\mathbf{Z} \in \mathbb{R}^{L \times H \times W}$, by convolving the CLIP visual features $\mathbf{F}$ with the object features $\mathbf{O}$ as follows:

$$\mathbf{Z}_{i,jk} = \sigma\left(\mathbf{O}_i^\top \mathbf{F}_{jk}\right), \tag{3}$$

where $\mathbf{Z}_{i,jk}$ is the relationship between the object feature $\mathbf{O}_i$ and the CLIP visual feature $\mathbf{F}_{jk}$, $\sigma$ denotes the sigmoid function. Similarly, we calculate the relationships between the segmentation features and the object features, denoted as $\mathbf{M} \in \mathbb{R}^{L \times H \times W}$, by convolving the segmentation features $\mathbf{B}$ with the object features $\mathbf{O}$ as follows:

$$\mathbf{M}_{i,jk} = \sigma\left(\mathbf{O}_i^\top \mathbf{B}_{jk}\right), \tag{4}$$

where $\mathbf{M}_{i,jk}$ is the relationship between the object feature $\mathbf{O}_i$ and the segmentation feature $\mathbf{B}_{jk}$. The relationships $\mathbf{M}$ also serve as the predicted object masks. Then, the final spatial relationships between the segmentation features and the CLIP visual features, denoted as $\mathbf{D} \in \mathbb{R}^{HW \times HW}$, are calculated as follows:

$$\mathbf{D}_{jk,mn} = \frac{1}{L} \sum_{i=1}^{L} \mathbf{M}_{i,jk} \cdot \mathbf{Z}_{i,mn}, \tag{5}$$

where $\mathbf{D}_{jk,mn}$ is the relationship between the segmentation feature $\mathbf{B}_{jk}$ and the CLIP visual feature $\mathbf{F}_{mn}$. The relationships $\mathbf{M}$ are supervised using object mask annotations, while the relationships $\mathbf{Z}$ are guided through the alignment of the remapped CLIP visual features with the segmentation features, as shown in Eq. (13). These supervisory signals enable accurate spatial remapping of the CLIP visual features $\mathbf{F}$.

We also apply another MLP to the object features $\mathbf{O}$ to generate object scores $\mathbf{S} \in \mathbb{R}^{L \times 2}$, indicating the background and foreground probabilities for each predicted object. The predicted objects are retained in the output only if their foreground probability surpasses a threshold $t$.

### 3.3 Spatial Remapping Module

Here, we describe the proposed spatial remapping module that remaps the CLIP visual features for better alignment with the object masks. The remapped CLIP visual features are then used in mask pooling for object classification.

As shown in Fig. 3 (c), after obtaining the spatial relationships $\mathbf{D} \in \mathbb{R}^{HW \times HW}$ between the segmentation features $\mathbf{B} \in \mathbb{R}^{H \times W \times C}$ and the CLIP visual features $\mathbf{F} \in \mathbb{R}^{H \times W \times C}$, we apply these relationships to remap the CLIP visual features $\mathbf{F}$ as follows:

$$
\begin{aligned}
\mathbf{V}_{jk} &= \frac{1}{HW} \sum_{m=1}^{H} \sum_{n=1}^{W} \mathbf{D}_{jk,mn} \cdot \mathbf{F}_{mn} \\
&= \frac{1}{HW} \sum_{m=1}^{H} \sum_{n=1}^{W} \left( \frac{1}{L} \sum_{i=1}^{L} \mathbf{M}_{i,jk} \cdot \mathbf{Z}_{i,mn} \right) \cdot \mathbf{F}_{mn} \\
&= \frac{1}{L} \sum_{i=1}^{L} \mathbf{M}_{i,jk} \left( \frac{1}{HW} \sum_{m=1}^{H} \sum_{n=1}^{W} \mathbf{Z}_{i,mn} \cdot \mathbf{F}_{mn} \right) \\
&= \frac{1}{L} \sum_{i=1}^{L} \mathbf{M}_{i,jk} \cdot \mathbf{E}_i,
\end{aligned}
\tag{6}
$$

where $\mathbf{V} \in \mathbb{R}^{H \times W \times C}$ is the remapped CLIP visual features, $\mathbf{E}_i \in \mathbb{R}^C$ is an intermediate embedding, $\mathbf{Z}_{i,mn}$ and $\mathbf{M}_{i,jk}$ are defined in Eq. (3) and Eq. (4), respectively.

For object classification, we obtain object embeddings $\mathbf{G} \in \mathbb{R}^{L \times C}$ by applying mask pooling to the remapped CLIP feature map $\mathbf{V}$. Next, we encode class names into text embeddings $\mathbf{T} \in \mathbb{R}^{K \times C}$ using a frozen CLIP text

encoder, where $K$ is the number of classes. Then, the classification probabilities $\mathbf{P} \in \mathbb{R}^{L \times K}$ are computed as follows:

$$\mathbf{P}_{ic} = \text{softmax}_c \left( \beta \cdot \frac{\mathbf{G}_i}{\|\mathbf{G}_i\|}^{\top} \frac{\mathbf{T}_c}{\|\mathbf{T}_c\|} \right), \tag{7}$$

where $\mathbf{G}_i \in \mathbb{R}^C$ is the embedding of the $i$-th object, $\mathbf{T}_c \in \mathbb{R}^C$ is the embedding of the $c$-th class, $\mathbf{P}_{ic}$ is the predicted probability that the $i$-th object belongs to class $c$, and $\beta$ is a scaling factor set to 100.

### 3.4 Training Loss

In this section, we introduce our loss function design, which includes object and alignment losses. To compute the object loss, we first calculate pairwise matching costs between the $L$ predicted objects and the $M$ ground-truth objects, including a foreground cost $\mathbf{L}^{fg}$, a classification cost $\mathbf{L}^{cls}$ and a segmentation cost $\mathbf{L}^{seg}$. $\mathbf{L}^{fg}$ is defined as follows:

$$\mathbf{L}_{ij}^{fg} = -\mathbf{S}_{j1}^* \log(\mathbf{S}_{i1}), \tag{8}$$

where $\mathbf{L}_{ij}^{fg}$ is the foreground cost between the $i$-th prediction and $j$-th ground-truth, $\mathbf{S}_{i1}$ is the foreground probability of the $i$-th prediction, $\mathbf{S}_{j1}^*$ is the foreground label of the $j$-th ground-truth, which is 1. $\mathbf{L}^{cls}$ is defined as follows:

$$\mathbf{L}_{ij}^{cls} = -\sum_{c=1}^{C} \mathbf{P}_{jc}^* \log(\mathbf{P}_{ic}), \tag{9}$$

where $\mathbf{L}_{ij}^{cls}$ is the classification cost between the $i$-th prediction and $j$-th ground-truth, $\mathbf{P}_{ic}$ is the probability that the $i$-th prediction belongs to the $c$-th class, $\mathbf{P}_{jc}^*$ is the class label of the $j$-th ground-truth. $\mathbf{L}^{seg}$ is defined as follows:

$$\mathbf{L}_{ij}^{seg} = \text{Focal}(\mathbf{M}_i, \mathbf{M}_j^*) + \text{Dice}(\mathbf{M}_i, \mathbf{M}_j^*), \tag{10}$$

where $\mathbf{L}_{ij}^{seg}$ is the segmentation cost between the $i$-th prediction and $j$-th ground-truth, $\mathbf{M}_i \in \mathbb{R}^{H \times W}$ is the $i$-th predicted object mask, $\mathbf{M}_j^* \in \mathbb{R}^{H \times W}$ is the $j$-th ground-truth object mask, Focal is focal loss and Dice is dice loss. Then, we can find an optimal injective function $z$ as follows:

$$\arg\min_z \sum_{j=1}^{M} \lambda_{fg} \mathbf{L}_{z(j),j}^{fg} + \lambda_{cls} \mathbf{L}_{z(j),j}^{cls} + \lambda_{seg} \mathbf{L}_{z(j),j}^{seg}, \tag{11}$$

where $z(j)$ is the index of the prediction assigned to the $j$-th ground-truth, $\lambda_{fg}$, $\lambda_{cls}$ and $\lambda_{seg}$ are the balance weights for $\mathbf{L}^{fg}$, $\mathbf{L}^{cls}$ and $\mathbf{L}^{seg}$, respectively. Then, the object loss $\mathbf{L}_{obj}$ is defined as follows:

$$\begin{aligned} &\frac{1}{M} \sum_{j=1}^{M} \left[ \lambda_{fg} \mathbf{L}_{z(j),j}^{fg} + \lambda_{cls} \mathbf{L}_{z(j),j}^{cls} + \lambda_{seg} \mathbf{L}_{z(j),j}^{seg} \right] \\ &- \frac{1}{L-M} \sum_{i \notin \mathbf{U}} \lambda_{bg} \log(\mathbf{S}_{i0}), \end{aligned} \tag{12}$$

where $\mathbf{U} = \{z(j)\}_{j=1}^{M}$ is the index set of the predictions matched to the ground-truths, $\mathbf{S}_{i0}$ is the background probability of the $i$-th prediction. Then, the alignment loss $\mathbf{L}_{align}$ is defined as follows:

$$\sum_{i=1}^{H} \sum_{j=1}^{W} \left[ \text{MSE}(\mathbf{V}_{ij}, \mathbf{B}_{ij}) + 1 - \text{Cosine}(\mathbf{V}_{ij}, \mathbf{B}_{ij}) \right], \tag{13}$$

where MSE is MSE loss, Cosine is consine similarity, The final loss is defined as follows:

$$loss = \mathbf{L}_{obj} + \lambda_{align} \mathbf{L}_{align}, \tag{14}$$

where $\lambda_{align}$ is the balanced weight for $\mathbf{L}_{align}$.

Table 1: Open-vocabulary panoptic segmentation performance on ADE20K, Mapillary Vistas and Cityscapes

| Method | ADE20K | | | | | Mapillary Vistas | | | | Cityscapes | | | | |
|---|---|---|---|---|---|---|---|---|---|---|---|---|---|---|
| | PQ | SQ | RQ | AP | mIoU | PQ | SQ | RQ | mIoU | PQ | SQ | RQ | AP | mIoU |
| MaskCLIP | 15.1 | 70.4 | 19.2 | 6.0 | 23.7 | - | - | - | - | - | - | - | - | - |
| FreeSeg | 16.3 | - | - | 6.5 | 24.6 | - | - | - | - | - | - | - | - | - |
| OPSNet | 19.0 | 52.4 | 23.0 | - | - | - | - | - | - | - | - | - | - | - |
| ODISE | 22.6 | - | - | 14.4 | 29.9 | 14.2 | 61.0 | 17.2 | - | 23.9 | 75.3 | 29.0 | - | - |
| ODISE (caption) | 23.4 | - | - | 13.9 | 28.7 | - | - | - | - | - | - | - | - | - |
| MasQCLIP | 23.3 | - | - | - | - | - | - | - | - | - | - | - | - | - |
| FC-CLIP | 26.8 | 71.5 | 32.2 | 16.8 | 34.1 | 18.2 | 57.7 | 22.9 | 27.9 | 44.0 | 75.4 | 53.6 | 26.8 | 56.2 |
| MAFT+ | 27.1 | 73.5 | 32.9 | - | - | - | - | - | - | - | - | - | - | - |
| **Ours** | **27.9** | **75.9** | **33.9** | **17.4** | **35.4** | **19.4** | **63.7** | **24.5** | **28.6** | **44.8** | **77.4** | **54.3** | **30.2** | **59.3** |

We report **panoptic quality** (PQ), **segmentation quality** (SQ), **recognition quality** (RQ), mask-level **average precision** (AP), and **mean IoU** (mIoU). PQ, SQ, RQ and mIoU are evaluated on both "thing" and "stuff" classes, while AP is computed on thing classes only. All values are percentages; the highest score in each column is highlighted in bold.

## 4 Experiments

### 4.1 Datasets and Metrics

Our model is trained on COCO panoptic dataset (Lin et al., 2014), including 118K training and 5K validation images across 80 "thing" and 53 "stuff" classes. Following prior works (Ghiasi et al., 2022; Xu et al., 2023a; Yu et al., 2023a), we evaluate our model on ADE20K (Zhou et al., 2017), Cityscapes (Cordts et al., 2016) and Mapillary Vistas (Neuhold et al., 2017) for open-vocabulary panoptic segmentation. ADE20K contains 2K validation images, with two versions: A-150 with 150 classes (100 "thing" and 50 "stuff") , and A-847 with 847 classes (only for semantic segmentation). Cityscapes contains 500 validation images across 8 "thing" and 11 "stuff" classes. Mapillary Vistas contains 2K validation images across 37 "thing" and 28 "stuff" classes. We also report open-vocabulary semantic segmentation results on those datasets as well as on PASCAL VOC (Everingham et al., 2010) and PASCAL Context (Mottaghi et al., 2014). PASCAL VOC contains 1.5K validation images, with two versions: PAS-21 with 20 foreground and 1 background classes, and PAS-20 without the background. PASCAL Context contains 5K validation images, with two versions: PC-59 with 59 classes, and PC-459 with 459 classes. Panoptic segmentation is evaluated using panoptic quality (PQ) (Kirillov et al., 2019), Average Precision (AP), and mean intersection-over-union (mIoU), while semantic segmentation is assessed with mIoU (Everingham et al., 2010). AP is calculated for "thing" classes, while PQ and mIoU include both "thing" and "stuff" classes.

### 4.2 Implementation Details

The CLIP model (Radford et al., 2021) we use is from OpenCLIP (Ilharco et al., 2021) with a ConvNeXt-Large backbone (Liu et al., 2022), and is pre-trained on LAION-2B dataset (Schuhmann et al., 2022). The segmentation network shares the convolutional backbone from the CLIP model, followed by a pixel decoder from (Yu et al., 2023a) to produce the segmentation feature map. The number of the object queries $L$ is set to 250. The input images are augmented with random horizontal flipping and random crop, and then resized and padded to $1024 \times 1024$. Optimization is done by AdamW (Loshchilov & Hutter, 2017) with betas of 0.9 and 0.999, and weight decay of 0.05. The batch size is set to 16. We train the model for 30 epochs. The learning rate is initialized at $2 \times 10^{-4}$, and multiplied by 0.1 after 20 and 25 epochs, respectively. The loss weights $\lambda_{fg}$, $\lambda_{bg}$, $\lambda_{cls}$, $\lambda_{seg}$ and $\lambda_{align}$ are set to 2, 0.2, 2, 5 and 5, respectively, to bring the losses to the same scale. The object threshold $t$ is set to 0.6.

### 4.3 Results

**Open-Vocabulary Panoptic Segmentation.** As shown in Table 1, when we compare our method with other state-of-the-art methods on the ADE20K dataset, our method achieves superior performance. Specifically, it outperforms MaskCLIP (Ding et al., 2023) by +12.8 PQ, +11.4 AP, and +11.7 mIoU; outperforms FreeSeg (Qin et al., 2023) (without using COCO-Stuff annotations) by +11.6 PQ, +10.9 AP, and +10.8

Table 2: Open-vocabulary semantic segmentation performance on ADE20K, PASCAL Context and PASCAL VOC

| Method | Training Dataset | mIoU | | | | | |
|---|---|---|---|---|---|---|---|
| | | A-847 | PC-459 | A-150 | PC-59 | PAS-21 | PAS-20 |
| SPNet (Xian et al., 2019) | Pascal VOC | - | - | - | 24.3 | 18.3 | - |
| ZS3Net (Bucher et al., 2019) | Pascal VOC | - | - | - | 19.4 | 38.3 | - |
| LSeg (Li et al., 2022a) | Pascal VOC | - | - | - | - | 47.4 | - |
| GroupViT (Xu et al., 2022a) | GCC + YFCC | 4.3 | 4.9 | 10.6 | 25.9 | 50.7 | 52.3 |
| SimBaseline (Xu et al., 2022b) | COCO Stuff | - | - | 15.3 | - | 74.5 | - |
| ZegFormer (Ding et al., 2022) | COCO Stuff | - | - | 16.4 | - | 73.3 | - |
| LSeg+ (Li et al., 2022a) | COCO Stuff | 3.8 | 7.8 | 18.0 | 46.5 | - | - |
| OVSeg (Liang et al., 2023) | COCO Stuff | 9.0 | 12.4 | 29.6 | 55.7 | - | 94.5 |
| SAN (Xu et al., 2023c) | COCO Stuff | 13.7 | 17.1 | 33.3 | **60.2** | - | 95.5 |
| EBSeg (Shan et al., 2024) | COCO Stuff | 13.7 | 21.0 | 32.8 | **60.2** | - | 96.4 |
| MAFT+ (Jiao et al., 2024) | COCO Stuff | 15.1 | **21.6** | 36.1 | 59.4 | - | **96.5** |
| USE+SAM (Wang et al., 2024) | COCO Stuff + VG | 13.4 | 15.0 | **37.1** | 58.0 | - | - |
| OpenSeg (Ghiasi et al., 2022) | COCO Panoptic + COCO Caption | 6.3 | 9.0 | 21.1 | 42.1 | - | - |
| ODISE (caption)(Xu et al., 2023a) | COCO Panoptic + COCO Caption | 11.0 | 13.8 | 28.7 | 55.3 | 82.7 | - |
| MaskCLIP (Ding et al., 2023) | COCO Panoptic | 8.2 | 10.0 | 23.7 | 45.9 | - | - |
| MasQCLIP (Xu et al., 2023d) | COCO Panoptic | 10.7 | 18.2 | 30.4 | 57.8 | - | - |
| ODISE (Xu et al., 2023a) | COCO Panoptic | 11.1 | 14.5 | 29.9 | 57.3 | **84.6** | - |
| FC-CLIP (Yu et al., 2023a) | COCO Panoptic | 14.8 | 18.2 | 34.1 | 58.4 | 81.8 | 95.4 |
| **Ours** | COCO Panoptic | **16.0** | 18.9 | 35.4 | 60.0 | 83.5 | 95.8 |

"A-847" and "A-150" refer to the 847- and 150-class splits of ADE20K, while "PC-459" and "PC-59" represent the 459- and 59-class splits of PASCAL Context. "PAS-21" and "PAS-20" denote PASCAL VOC with and without the background class, respectively. The evaluation metric is mIoU, and bold numbers indicate the best performance.

mIoU; outperforms OPSNet (Chen et al., 2023b) by +8.9 PQ, outperforms ODISE (Xu et al., 2023a) by +5.3 PQ, +3.0 AP, and +5.5 mIoU; outperforms ODISE with caption annotations for training by +4.5 PQ, +3.5 AP, and +6.7 mIoU; outperforms MasQCLIP (Xu et al., 2023d) by +4.6 PQ, outperforms FC-CLIP (Yu et al., 2023a) by +1.1 PQ, +0.6 AP, and +1.3 mIoU; and outperforms MAFT+ (Jiao et al., 2024) by +0.8 PQ. For street-view datasets, our method continues to outperform concurrent works. It surpasses ODISE by +5.2 PQ on Mapillary Vistas and by +20.9 PQ on Cityscapes. Compared to FC-CLIP, it achieves improvements of +1.2 PQ and +0.7 mIoU on Mapillary Vistas, and +0.8 PQ and +3.1 mIoU on Cityscapes. This further demonstrates the robustness of our method.

We also provide visualization of comparison results on ADE20K in Fig. 4, with more examples in the supplementary materials. The comparison method, FC-CLIP (Yu et al., 2023a), ensembles object embeddings from CLIP visual features with those from newly learned visual features for zero-shot object classification. In contrast, our approach spatially remaps CLIP visual features to better align with object masks, enabling accurate classification via mask pooling on the remapped features. Both models are trained on the COCO panoptic dataset and evaluated zero-shot on the ADE20K validation set. Across diverse scenes, our visualizations show that FC-CLIP often misses rare categories, confuses adjacent "thing" and "stuff" regions when object cues are weak, and exhibits semantic drift on long-tail classes. In contrast, our spatial remapping identifies rare categories more reliably and produces steadier label assignments under heavy occlusion or clutter. We also observe fewer cases where masks are correct but labels are inconsistent, indicating that remapping improves feature-to-mask correspondence before classification. Overall, the spatial remapping module better aligns CLIP visual features with object masks, yielding more accurate zero-shot classification and improved open-vocabulary segmentation.

**Open-Vocabulary Semantic Segmentation.** Although our model is trained only on the COCO panoptic dataset, it performs well on open-vocabulary semantic segmentation tasks. As shown in Table 2, when comparing our method with other state-of-the-art methods also trained only on the COCO panoptic dataset, our method consistently outperforms them. Specifically, evaluated on the A-847, PC-459, A-150, and PC-59 datasets, it surpasses MaskCLIP by +7.8, +8.9, +11.7, and +14.1 mIoU, respectively; outperforms MasQCLIP by +5.3, +0.7, +5.0, and +2.2 mIoU, respectively; outperforms ODISE by +4.9, +4.4, +5.5, and +2.7 mIoU; and exceeds FC-CLIP by +1.2, +0.7, +1.3, and +1.6 mIoU on the respective datasets. Compared

Table 3: Ablation Studies on ADE20K and Cityscapes for Open-Vocabulary Panoptic Segmentation

| Remapping | Relation | Supervision | ADE20K | | | | Cityscapes | | | |
|---|---|---|---|---|---|---|---|---|---|---|
| | | | PQ | AP | mIoU | FPS | PQ | AP | mIoU | FPS |
| ✘ | ✘ | ✘ | 20.3 | 12.3 | 28.2 | 2.78 | 27.4 | 22.0 | 37.3 | 1.52 |
| ✔ | Pixel-Level | ✘ | 25.2 | 15.2 | 31.5 | 0.64 | 40.8 | 26.1 | 54.2 | 0.24 |
| ✔ | Object-Level | ✘ | 24.6 | 14.2 | 30.2 | 2.13 | 40.3 | 24.0 | 53.2 | 1.25 |
| ✔ | Object-Level | ✔ | **27.9** | **17.4** | **35.4** | 2.13 | **44.8** | **30.2** | **59.3** | 1.25 |

We evaluate the effect of each proposed component – spatial remapping, spatial relation granularity (pixel- vs object-level), and relation supervision. Metrics are reported on the ADE20K and Cityscapes test sets using PQ, AP, mIoU, and inference speed (FPS). "✔" indicates the component is enabled and "✘" indicates it is disabled. The relation supervision is only applicable to object-level relations. Bold numbers mark the best performance; the full model (spatial remapping, object-level relations, relation supervision) achieves the largest accuracy gains while retaining real-time efficiency.

to methods utilizing caption annotations, our approach still shows advantages. For instance, it surpasses OpenSeg by +9.7, +9.9, +14.3, and +17.9 mIoU, surpasses ODISE (caption) by +5.0, +5.1, +6.7, and +4.7 mIoU on A-847, PC-459, A-150, and PC-59, respectively.

Even when compared to methods trained on COCO Stuff—which includes 38 more classes than COCO Panoptic—our method demonstrates comparable or superior performance. Specifically, on the A-847, PC-459, A-150, and PC-59 datasets, our method outperforms LSeg+ by +12.2, +11.1, +17.4, and +13.5 mIoU, respectively; outperforms OVSeg by +7.0, +6.5, +5.8, and +4.3 mIoU, respectively. On A-847, PC-459 and A-150, our method outperforms SAN by +2.3, +1.8 and +2.1 mIoU, respectively. On A-847 and A-150, our method exceeds EBSeg by +2.3 and +2.6 mIoU, respectively. On A-847 and PC-59, our method exceeds MAFT+ by +0.9 and +0.6 mIoU, respectively. On A-847 and PC-459, our method exceeds USE+SAM by +2.6 and +3.9 mIoU, respectively. These results highlight the robust zero-shot object classification ability of our model. This is because our spatial remapping module effectively aligns CLIP visual features with object masks without the need for newly learned visual features, thereby preserving CLIP's zero-shot image-text alignment capability as much as possible.

### 4.4 Ablation Studies

**Influence of Spatial Remapping.** We compare the panoptic segmentation performance of models with and without the proposed spatial remapping of CLIP visual features on the ADE20K and Cityscapes datasets. For the model without spatial remapping, we directly obtain object embeddings by pooling CLIP visual features using the generated object masks, as shown in Fig. 2(a). As shown in Table 3, the model with the spatial remapping module outperforms the one without it, achieving improvements of +7.6 PQ, +5.1 AP, and +7.2 mIoU on ADE20K; and +17.4 PQ, +8.2 AP, and +22.0 mIoU on Cityscapes. These results highlight the significance of spatially remapping CLIP visual features for better alignment with object masks. This alignment enables the object embeddings, derived from mask pooling on the remapped CLIP visual features, to correctly match the text embeddings, thereby facilitating accurate object classification. Compared with other methods fine-tuning CLIP features (e.g., FC-CLIP), the proposed spatial remapping of CLIP features can better preserve CLIP's zero-shot image-text alignment ability learned from internet-scale image-text pair dataset as much as possible.

**Supervision of Spatial Relationships.** Using the spatial remapping module shown in Fig. 3, we compare panoptic segmentation performance on the ADE20K and Cityscapes datasets, with and without supervising the object-level spatial relationships between CLIP visual features and segmentation features using object mask annotations. When these spatial relationships are not supervised, the predicted object masks are obtained from an alternative mask prediction head rather than the object-level spatial relationship module. As shown in Table 3, supervising the object-level spatial relationships with object mask annotations results in performance gains of +3.3 PQ, +3.2 AP, and +5.2 mIoU on ADE20K, and +4.5 PQ, +6.2 AP, and +6.1 mIoU on Cityscapes compared to not supervising them. These results demonstrate that such supervision is essential for learning accurate spatial relationships between CLIP visual features and segmentation features.

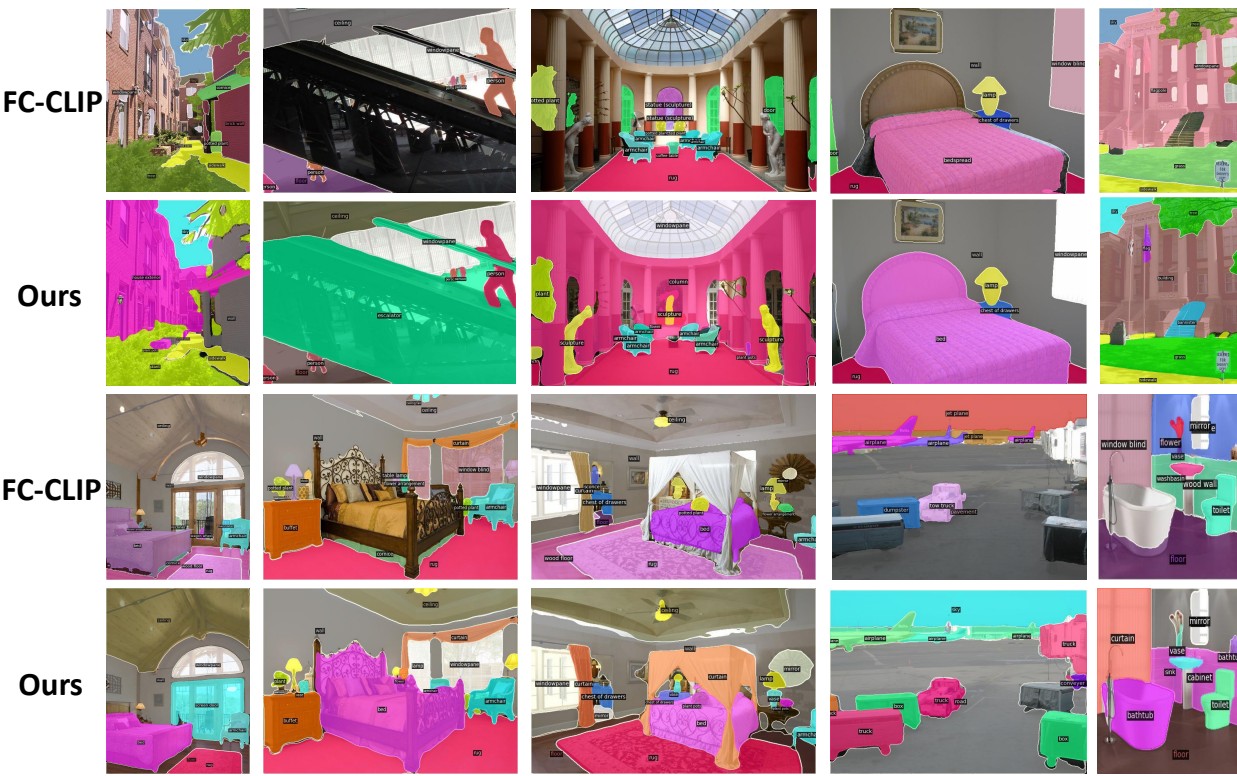

Figure 4: Visualization of panoptic segmentation on ADE20K. The comparison method, FC-CLIP Yu et al. (2023a), fine-tunes object embeddings extracted from the CLIP feature map with mask pooling, by integrating newly learned embeddings for zero-shot object classification. In contrast, our approach recombines CLIP features for better alignment with object masks, before applying mask pooling to the CLIP feature map to extract object embeddings. Both models are trained on the COCO panoptic training set and undergo zero-shot evaluation on the ADE20K validation set.

This enables us to remap CLIP features for better alignment with object masks, thereby improving object classification accuracy.

Table 4: Open-vocabulary panoptic segmentation performance on COCO validation set.

| Method | COCO (i.i.d.) | | |
|---|---|---|---|
| | **PQ** | **AP** | **mIoU** |
| MaskCLIP (Ding et al., 2023) | - | - | - |
| FreeSeg (Qin et al., 2023) | - | - | - |
| ODISE (Xu et al., 2023a) | **55.4** | **46.0** | **65.2** |
| ODISE (caption)(Xu et al., 2023a) | 45.6 | 38.4 | 52.4 |
| MasQCLIP (Xu et al., 2023d) | - | - | - |
| FC-CLIP (Yu et al., 2023a) | 54.4 | 44.6 | 63.7 |
| **Ours** | 54.9 | 44.8 | 64.9 |

The corresponding training set is utilized to train our model. The evaluation metrics are PQ, AP and mIoU.

**Object-Level Relation vs. Pairwise Relation.** We investigate the influence of modeling the spatial relationships between CLIP visual features and segmentation features at the object level versus in pairs. In the object-level approach, we use object features as intermediaries to establish the relationships between CLIP visual features and segmentation features, as depicted in Fig. 3(b). In the pairwise approach, the relationships are established through direct pairwise comparisons between CLIP visual features and segmen-

tation features. As shown in Table 3, modeling the spatial relationships at the object level outperforms the pairwise modeling, showing gains of +2.7 PQ, +2.2 AP, and +3.9 mIoU on ADE20K, and +4.0 PQ, +4.1 AP, and +5.1 mIoU on Cityscapes. These results demonstrate that spatial relationships modeled at the object level are more accurate than those modeled pairwise. This is because object-level modeling enables supervision of the relationships using object mask annotations. Furthermore, modeling spatial relationships at the object level improves the FPS by 230%, demonstrating its efficiency. This is because modeling spatial relationships at the object level greatly reduce the computational and memory cost compared to modeling spatial relationships pairwise.

**Independent and Identically Distributed (i.i.d.) Results** We also evaluate the panoptic segmentation performance of our model on the COCO validation set, for which the corresponding training set is utilized to train our model. As shown in Tab. 4, when comparing our method with other state-of-the-art methods, our method achieves comparable performance; e.g., with +9.3 PQ, +6.4 AP, and +12.5 mIoU higher than ODISE (caption), and +0.5 PQ, +0.2 AP, and +1.2 mIoU higher than FC-CLIP. The comparison results show that our method can still achieve good object segmentation and recognition performance in iid. distribution, demonstrating the effectiveness of the proposed spatial remapping module and spatial relationship modeling, which accurately align CLIP visual features with object masks. Although our method exhibits marginally lower accuracy compared to ODISE, this is intentional to enhance its generalization ability across various datasets. Since we do not finetune the CLIP visual encoder and only remap its features at the spatial level, we better preserve CLIP's zero-shot image–text alignment ability while avoiding overfitting.

## 5 Conclusion

We presented a simple yet effective spatial remapping module that aligns CLIP visual features with object masks before classification, yielding consistent gains for open-vocabulary segmentation. By leveraging the spatial relationships between CLIP visual features and segmentation features, this module remaps the CLIP visual features to better align with object masks, enabling accurate object classification through mask pooling. Trained only on COCO panoptic and evaluated strictly zero-shot on diverse benchmarks (e.g., ADE20K, Cityscapes, Mapillary Vistas, PASCAL VOC, and PASCAL Context), our approach preserves CLIP's language–vision alignment while correcting mask-level misalignment, enabling accurate mask-pooled recognition without additional category-specific fine-tuning. Qualitative comparisons indicate crisper boundaries, improved part–whole delineation, and fewer label drifts on long-tail concepts; quantitative results further confirm robust improvements over representative CLIP-based baselines such as FC-CLIP. Additionally, modeling spatial relationships between CLIP visual features and segmentation features at the object level improves computational efficiency by shifting from dense, pixel-wise interactions to sparse, object-conditioned relationship modeling, which yields lower memory traffic and better real time performance in practice.

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
