# OpenReview forum: "Aligning CLIP Visual Features with Object Masks via Spatial Remapping for Open-Vocabulary Segmentation"
_TMLR — Under review for TMLR_

### Review · Reviewer_s8Lk · 2026-06-01

**Summary Of Contributions:**

This paper proposes an object-level spatial relationship module and a spatial remapping module for CLIP tailored to open-vocabulary panoptic segmentation, which transforms CLIP visual features to better align with object masks before mask pooling, thereby eliminating the need to ensemble with poorly generalizing newly learned segmentation features. Trained solely on COCO panoptic, the method achieves exceptional zero-shot results across various datasets.

**Audience:**

Yes

**Audience Explanation:**

This work tackles a timely and significant challenge in open-vocabulary segmentation, appealing to researchers working on CLIP-based segmentation, zero-shot recognition, and open-world perception, along with practitioners in autonomous driving, robotics, and augmented reality who require models capable of generalizing to unseen categories. The proposed spatial remapping approach is straightforward, computationally efficient, and retains CLIP's original zero-shot capability without requiring full backbone fine-tuning, rendering it highly suitable for real-world deployment.

**Broader Impact Concerns:**

This paper presents a method to improve open-vocabulary segmentation with positive societal implications for autonomous driving, robotics, and so on. However, potential risks include the preservation or amplification of biases inherited from CLIP's internet-scale pretraining data, and the environmental cost of training and deploying large foundation models—concerns the authors should explicitly acknowledge in a dedicated broader impact discussion.

**Claims And Evidence:**

Yes

**Claims Explanation:**

1.	The article has a clear structure and is well-organized. Figure 1 visually shows the motivation of the proposed method. Figure 2 clearly shows the difference between the proposed pipeline and other pipelines, and Figure 3 clearly illustrates the overview of the proposed framework.
2.	The paper includes well-designed ablations that isolate the contribution of each component (spatial remapping, object-level relations, supervision), clearly demonstrating their individual impact on performance and efficiency.
3.	The method is trained solely on COCO panoptic yet evaluated across diverse datasets, achieving excellent performance.

**Requested Changes:**

1.	The method is evaluated only with ConvNeXt-Large CLIP. Given the field's rapid shift toward ViT-based vision transformers, the lack of experiments or discussion on whether spatial remapping transfers across CLIP architectures weakens generalizability claims.
2.	As shown in Table 4, the proposed method exhibits lower accuracy compared to ODISE. If you finetune the CLIP visual encoder, will the accuracy be improved to surpass that of ODISE?
3.	The paper describes 2.13 FPS as "real-time efficiency," which is misleading—2 FPS is far below real-time video standards (30 FPS). While the relative speedup over pixel-level alternatives is valid, the absolute framing overstates practical deployability.

---

### Review · Reviewer_KKXM · 2026-06-12

**Summary Of Contributions:**

The authors address the challenge of open-vocabulary panoptic segmentation, specifically the discrepancy between CLIP’s visual features and object masks when object-level embeddings are derived through mask pooling. They argue that directly pooling frozen CLIP features over predicted masks can yield suboptimal object embeddings due to CLIP’s training on image-level supervision rather than mask-level alignment. To mitigate this issue, the paper proposes a spatial remapping module that reorganizes or remaps CLIP visual features to enhance their alignment with object masks prior to mask pooling and text-image matching.

A pivotal technical contribution lies in the development of an object-level spatial relationship module. Rather than computing dense pairwise relationships between all CLIP and segmentation feature locations, the method employs object queries or features as intermediaries. These object-level relationships are supervised with mask annotations and subsequently utilized to remap CLIP features. This design aims to optimize computational efficiency while simultaneously improving the quality of the CLIP-mask alignment. The final object embeddings are obtained through mask pooling over the remapped CLIP features and classified using frozen CLIP text embeddings, thereby preserving CLIP’s zero-shot recognition capabilities.

Empirically, the paper presents compelling results on open-vocabulary panoptic segmentation and semantic segmentation benchmarks. The model undergoes training on COCO panoptic data and is subsequently evaluated zero-shot on datasets such as ADE20K, Cityscapes, Mapillary Vistas, PASCAL VOC, and PASCAL Context. The reported results demonstrate consistent improvements over several prior CLIP-based open-vocabulary segmentation methods, including FC-CLIP, with ablations supporting the significance of spatial remapping, object-level relation modeling, and mask supervision.

The primary strengths of the paper lie in its clear motivation, straightforward and intuitive formulation, and robust empirical performance across multiple benchmarks. The proposed object-level remapping mechanism is also appealing because it directly addresses a well-known weakness of CLIP-based segmentation pipelines: the spatial misalignment between CLIP features and object masks. The ablation studies are valuable and provide evidence that the proposed components contribute significantly to the final performance.

The main weaknesses are that the novelty relative to prior CLIP feature adaptation, mask alignment, and object-query-based segmentation methods could be articulated more explicitly. The paper would also benefit from a deeper analysis of what the remapping operation preserves or alters in CLIP’s feature space, as well as a more thorough discussion of potential failure cases and computational trade-offs. Some claims regarding preserving CLIP’s zero-shot alignment are plausible but would be strengthened with more direct evidence beyond downstream benchmark gains.

**Additional Comments:**

NA

**Audience:**

Yes

**Audience Explanation:**

The findings would likely be of interest to at least some TMLR readers, especially those working on vision-language models, open-vocabulary recognition, and segmentation. The paper studies an important limitation of CLIP-based segmentation pipelines, namely that CLIP features are useful for zero-shot classification but are not naturally aligned with object masks. The proposed spatial remapping approach is a simple and relevant idea for improving mask-level recognition while retaining the benefits of frozen CLIP representations. Since the paper reports consistent gains across several open-vocabulary panoptic and semantic segmentation benchmarks, the results should be useful to researchers studying how to adapt large vision-language models for dense prediction tasks.

**Claims And Evidence:**

Yes

**Claims Explanation:**

The main claims are supported by reasonably clear and convincing evidence. The paper identifies a concrete problem in CLIP-based open-vocabulary segmentation: mask-pooled CLIP features are not well aligned with object masks, and it proposes a spatial remapping module to address this issue. The empirical results show consistent improvements over prior methods on open-vocabulary panoptic segmentation benchmarks such as ADE20K, Mapillary Vistas, and Cityscapes, as well as strong semantic segmentation results on ADE20K, PASCAL Context, and PASCAL VOC. The ablation studies further support the core claims by showing that spatial remapping, object-level relation modeling, and supervision of the spatial relationships each contribute to performance. However, some broader claims, especially that the method better preserves CLIP’s zero-shot image-text alignment, would be more convincing with direct feature-space analysis, category-level breakdowns, and more detailed failure-case studies. Overall, the evidence supports the central contribution, though the analysis could be deeper.

**Requested Changes:**

- Provide a sharper discussion of novelty relative to closely related CLIP-based open-vocabulary segmentation methods, especially FC-CLIP, MaskCLIP, MasQCLIP, and ODISE. The paper should make clearer what is technically new beyond adapting or realigning CLIP features for mask-level classification.

- Add more direct evidence for the claim that the proposed remapping preserves CLIP’s zero-shot image-text alignment. The current benchmark results support this indirectly, but feature-space analysis, CLIP-text similarity analysis, or category-level generalization results would make the claim more convincing.

- Clarify the computational trade-offs of the proposed module. The FPS ablation is useful, but the paper should also report parameter count, FLOPs, or memory usage, and ensure comparisons are made under the same hardware and input resolution.

- Include more detailed breakdowns of results, such as thing versus stuff, rare versus frequent categories, small versus large objects, and categories that are semantically close to or far from COCO classes.

- Add a more informative failure-case analysis. The paper should discuss when spatial remapping does not help, for example on small objects, thin structures, occlusion, cluttered scenes, or ambiguous stuff classes.

- Improve clarity and reproducibility by expanding implementation details, including text prompt templates, inference-time post-processing, FPS measurement setup, and whether baseline numbers are reproduced or taken from prior work.

---

### Review · Reviewer_oPi7 · 2026-06-16

**Summary Of Contributions:**

Summary:
The paper tackles the open-vocabulary segmentation problem with a focus on open-vocabulary panoptic segmentation. The core observation is that CLIP features are trained for image-level text-image alignment, which might not be well aligned with object-level masks. Hence, directly applying mask pooling on CLIP features can produce poor object embeddings for category classification.  To solve this issue, the paper proposes a spatial remapping module that reorganizes CLIP visual features before mask pooling, so that the remapped features better align with predicted object masks while still preserving CLIP’s pretrained semantic alignment.

Strengths:
1. The paper targets a real limitation of CLIP-based open-vocabulary segmentation and proposes a simple yet effective feature-remapping strategy.
2.  The proposed spatial remapping module is well motivated: it keeps CLIP’s zero-shot semantic space while making its feature map more mask-aligned before pooling.
3.  The ablation results are convincing: spatial remapping gives large gains over direct CLIP pooling, and object-level relationship supervision further improves performance.

Weakness:
1.   The gains over the strongest baselines are relatively small. While the improvement over direct CLIP pooling is large, the improvement over FC-CLIP and other recent methods is much smaller.
2.   The method may depend heavily on mask quality. If the object masks are inaccurate, it is unclear whether spatial remapping can still improve object classification.
3.   The paper does not compare with very recent segmentation foundation models such as SAM3.

**Audience:**

Yes

**Audience Explanation:**

Yes. I believe at least some individuals in TMLR’s audience would be interested in this paper. The proposed spatial remapping approach provides a useful perspective on how to better adapt pretrained vision-language features for dense prediction while preserving their zero-shot recognition ability. Although the method is somewhat incremental relative to recent CLIP-based segmentation pipelines, the findings are still relevant and potentially useful to this research community.

**Broader Impact Concerns:**

I do not see major ethical concerns that would prevent publication.

**Claims And Evidence:**

Yes

**Claims Explanation:**

Overall, the main claims are supported by the experimental evidence. The paper evaluates the proposed method on multiple open-vocabulary panoptic and semantic segmentation benchmarks, and the results show consistent improvements over several relevant baselines. However, there are few missing critical pieces of evidence. For example, the paper could provide a more detailed analysis on sensitivity to mask quality and discussion with more recent segmentation foundation models like sam3.

**Requested Changes:**

1. The paper should provide a clearer novelty analysis over prior CLIP-region alignment methods. The current presentations makes the motivation clear, but the novelty of methods such as FC-CLIP, OVSeg, MAFT-style could be explained more explicitly.
2. It would be useful to include a comparison with recent segmentation foundation models such as SAM3. The visual encoder of SAM3 is actually CLIP-style pre-trained architecture: perception encoder[1].
3. The authors could include more analysis on the dependency of the method on mask quality. Since the spatial remapping relies on object-level relationships and predicted masks, it would be helpful to evaluate whether the method remains effective when mask proposals are noisy or incomplete.

[1] Bolya, Daniel, et al. "Perception encoder: The best visual embeddings are not at the output of the network." Advances in Neural Information Processing Systems 38 (2026): 60884-60937.